# Melatonin Action in Type 2 Diabetic Parotid Gland and Dental Pulp: In Vitro and Bioinformatic Findings

**DOI:** 10.3390/ijerph20186727

**Published:** 2023-09-07

**Authors:** Milena Barać, Milan Petrović, Nina Petrović, Nataša Nikolić-Jakoba, Zoran Aleksić, Lidija Todorović, Nataša Petrović-Stanojević, Marina Anđelić-Jelić, Aleksandar Davidović, Jelena Milašin, Jelena Roganović

**Affiliations:** 1Department of Pharmacology in Dentistry, Faculty of Dental Medicine, University of Belgrade, 11000 Belgrade, Serbia; milenacimbaljevic@gmail.com; 2Clinic for Maxillofacial Surgery, Faculty of Dental Medicine, University of Belgrade, 11000 Belgrade, Serbia; milan.petrovic@stomf.bg.ac.rs; 3Vinča Institute of Nuclear Sciences, National Institute of the Republic of Serbia, University of Belgrade, 11000 Belgrade, Serbia; dragoninspiration@yahoo.com (N.P.); lidijat@vinca.rs (L.T.); 4Department of Periodontology, Faculty of Dental Medicine, University of Belgrade, 11000 Belgrade, Serbia; natasa.nikolic.jakoba@stomf.bg.ac.rs (N.N.-J.); dr.zoran.aleksic@gmail.com (Z.A.); 5Zvezdara University Medical Center, University of Belgrade, 11000 Belgrade, Serbia; natasa.petrovic@stomf.bg.ac.rs (N.P.-S.); drmajelic@gmail.com (M.A.-J.); davidovicalex@hotmail.com (A.D.); 6Department of Human Genetics, Faculty of Dental Medicine, University of Belgrade, 11000 Belgrade, Serbia; jelena.milasin@stomf.bg.ac.rs

**Keywords:** diabetes mellitus, type 2, poor oral health, parotid gland, dental pulp, melatonin, glial cell line-derived neurotrophic factor, inducible nitric oxide synthase, superoxide dismutase

## Abstract

Type 2 diabetes mellitus (T2DM) is associated with functional deterioration of the salivary gland and dental pulp, related to oxidative stress. The aim was to integrate experimental and bioinformatic findings to analyze the cellular mechanism of melatonin (MEL) action in the human parotid gland and dental pulp in diabetes. Human parotid gland tissue was obtained from 16 non-diabetic and 16 diabetic participants, as well as human dental pulp from 15 non-diabetic and 15 diabetic participants. In human non-diabetic and diabetic parotid gland cells (hPGCs) as well as in dental pulp cells (hDPCs), cultured in hyper- and normoglycemic conditions, glial cell line-derived neurotrophic factor (GDNF), MEL, inducible nitric oxide synthase (iNOS) protein expression, and superoxide dismutase (SOD) activity were measured by enzyme-linked immunosorbent assay (ELISA) and spectrophotometrically. Bioinformatic analysis was performed using ShinyGO (v.0.75) application. Diabetic participants had increased GDNF and decreased MEL in parotid (*p* < 0.01) and dental pulp (*p* < 0.05) tissues, associated with increased iNOS and SOD activity. Normoglycemic hDPCs and non-diabetic hPGCs treated with 0.1 mM MEL had increased GDNF (*p* < 0.05), while hyperglycemic hDPCs treated with 1 mM MEL showed a decrease in up-regulated GDNF (*p* < 0.05). Enrichment analyses showed interference with stress and ATF/CREB signaling. MEL induced the stress-protective mechanism in hyperglycemic hDPCs and diabetic hPGCs, suggesting MEL could be beneficial for diabetes-associated disturbances in oral tissues.

## 1. Introduction

It is well known that diabetes mellitus (DM) is a progressive metabolic disease caused by a relative or absolute deficit in the secretion of insulin and/or deterioration in its signaling, thus resulting in the appearance of hyperglycemia and disturbed metabolism of carbohydrates, fats and proteins. Due to the high rate of prevalence and distribution, along with the occurrence of severe accompanying complications, DM represents a significant global public health and economic burden.

In the Report of the Institute for Public Health of Serbia, “Dr. Milan Jovanović Batut” from 2020, it was announced that in the Republic of Serbia, seven hundred thousand people suffer from DM and that in 2040, every fifth citizen of Serbia will have DM. In Serbia, about 3000 people die from DM and its complications annually, and the number of deaths is probably even higher due to the high prevalence of undiagnosed cases and due to inaccurate recording of the cause of death (Available online: www.batut.org.rs/download/publikacije/Dijabetes2020.pdf, accessed on 15 April 2023). Type 2 diabetes mellitus (T2DM) induces deteriorating functional status of oral tissues and leads to disturbed oral homeostasis [1,2], with oxidative stress as one of the key mechanisms involved [2,3]. Salivary glands [4,5], dental pulp [6], periodontal tissues [7,8] and oral mucosa [9] are affected by the changes. Therefore, major accompanying features of T2DM in the oral cavity are hyposalivation and xerostomia, difficulty in chewing, swallowing and speech, altered sense of taste (dysgeusia), more frequent tooth loss due to a higher prevalence of periodontitis and dental caries, occurrence of infections, as well as delayed wound healing [9,10].

In diabetes, oxidative–nitrosative stress occurs due to persistent hyperglycemia, which induces a sequence of events that generate excessive production of reactive oxygen (ROS) and nitrogen species (RNS) [11]. Oxidative stress and inflammation are closely interconnected processes in diabetes, leading to endothelial dysfunction and damage to cellular components and signaling, resulting in oral tissue function deterioration in diabetes [3,12,13,14,15] (Figure 1).

As oral homeostasis is closely related to overall psycho-physical health, as well as to the maintenance of optimal glycemia, new therapeutic strategies targeting diabetes-induced oxidative stress and establishing oral homeostasis in patients with T2DM is of particular importance [16]. 

Melatonin (MEL) exhibits a strong antioxidant and anti-inflammatory effect, and its presence in the oral cavity helps maintain oral homeostasis [17,18]. The origin of MEL in saliva could be explained by the presence of a fraction of unbound MEL from the blood plasma, which, through the process of passive diffusion in the salivary glands, reaches the saliva. However, as shown in animal and human studies, the presence of enzymes that participate in the synthesis of MEL were found in the ductal cells as well as in the oral mucosa of the tongue, indicating a local synthesis of MEL in the excretory ducts of the salivary glands and in the oral cavity [19]. In addition, the existence of melatonin secretory granules and vesicles, in acinar and ductal cells, indicates the ability of salivary glands to deposit MEL [19]. Thus, MEL acts as an anti-inflammatory and antioxidant molecule in oral tissues [17,18,19] but also shows a protective role in a high glucose environment in vitro [3]. However, before considering MEL as a treatment for diabetic complications, cellular mechanisms of MEL action should be clarified in order to develop an effective and safe treatment. The aim of the present manuscript was to integrate in vitro and bioinformatic findings to analyze the cellular mechanism of MEL action in the parotid gland and dental pulp in diabetes.

## 2. Materials and Methods

### 2.1. Study Participants and Tissue Samples Collection

Prior to performing the research, the study protocol was approved by the Ethical Committee of the Faculty of Dental Medicine, University of Belgrade (approval Nº 36/2), and the research was carried out in accordance with the criteria of the Helsinki declaration (World Medical Association). Written informed consent was provided by all study participants. Human parotid gland tissue was obtained from a total of 32 participants (16 healthy and 16 diabetic), which underwent surgical removal of benign parotid tumors at the Clinic for Maxillofacial Surgery, Faculty of Dental Medicine. Surgically removed parotid tissue without histopathological signs of alterations was used in the study, and samples were transported in medium and then stored. Taking into account the circadian rhythm of MEL secretion, tissue samples were collected at the same time, in the period from 8:00 a.m. to 11:00 a.m. The samples were then transported in Krebs-Ringer’s bicarbonate nutrient solution, which contains NaCl 118.3 mM, KCl 4.7 mM, CaCl_2_ 2.5 mM, MgSO_4_ 1.2 mM, KH_2_PO_4_ 1.2 mM, NaHCO_3_ 25.0 mM and glucose 11.1 mM, to the laboratory where they were immediately frozen and stored at −70 °C for future experimental analysis. Dental pulp tissue was obtained from 30 participants (15 healthy and 15 diabetic) recruited at the Clinic for Restorative Dentistry and Endodontics, Faculty of Dental Medicine, due to pre-prosthetic endodontic management, and by the protocol previously used [3]. Namely, dental pulp tissue was obtained from intact teeth, after vital dental pulp extirpation, which was indicated due to pre-prosthetic preparation. The vitality of dental pulp tissue was checked clinically using an electro-apparatus and based on retro-alveolar radiographs. Teeth with an optimal vitality test and without signs of dental pulp and periodontium disease on the radiograph were included. Due to the circadian rhythm of MEL secretion, tissue samples were always collected between 10:00 and 12:00. Sampling and further treatment of the tooth were carried out according to a standard procedure by a specialist in dental diseases and endodontics. For local anesthesia, 2% lidocaine-chloride with adrenaline 1:100,000 was used (lidocaine 2%-adrenalin; Galenika, Belgrade, Serbia). The teeth were first disinfected with 70% ethanol and then with 0.2% chlorhexidine (Curasept 220; Curaden International AG, Kriens, Switzerland). After the trepanation, the pulp chamber was washed with a saline solution, and then the dental pulp tissue extirpation procedure occurred. Dental pulp tissue samples were immediately frozen in liquid nitrogen and transported to the laboratory where they were stored at −70 °C. Using the data from the study of Cutando et al. [19] with a sample size of a total of 30 participants with 15 participants per group, for a study power of 80% and α = 0.05, was calculated by statistical software G*Power. Participants who met the following inclusion criteria were enrolled in the study: ≥18 years of age, self-reported good general health, except for diabetic participants, who were on oral antidiabetic therapy, with well-controlled T2DM (HbA1c < 7%), while exclusion criteria were: presence of any other systemic disease; the use of immunosuppressants, antibiotics or anti-inflammatory drugs in the last month; pregnancy or lactation. 

### 2.2. Enzyme-Linked Immunosorbent Assay (ELISA)

Prior to the analyses, tissue was weighed and, in order to isolate tissue proteins, subjected to mechanical homogenization in phosphate buffered saline. All chemicals used in the study were obtained from Sigma-Aldrich (St. Louis, MO, USA) unless otherwise specified. After tissue disruption, samples were centrifuged at 10,000× *g*/10 min and then, in collected tissue supernatants, MEL (pg/mL), glial cell line-derived neurotrophic factor (GDNF) (pg/mL), and inducible nitric oxide synthase (iNOS) (ng/mL) proteins were determined using the following commercial ELISA kits in accordance with manufacturer’s protocols: Human Melatonin ELISA kit (Cusabio Biotech, Wuhan, China), Human GDNF ELISA kit (Sigma-Aldrich, St. Louis, MO, USA), Human iNOS ELISA Kit (Biopeony, Beijing, China). Total superoxide dismutase (SOD) activity (%) was assayed by spectrophotometric commercial SOD assay (SOD Determination Kit, Sigma Aldrich, St. Louis, MO, USA) following the manufacturer’s protocols.

### 2.3. Human Dental Pulp Cells (hDPCs) Culture

Dental pulp cells were isolated from human intact wisdom teeth with complete root formation obtained from two healthy participants after tooth extraction due to orthodontic reasons, at the Faculty of Dental Medicine, University of Belgrade. Explant outgrowth method was used for hDPCs isolation, as described previously [3]. Pulp fragments were cultured in Dulbecco’s modified Eagle’s medium supplemented with 10% fetal bovine serum, penicillin-G (100 units/mL), streptomycin (100 μg/mL) and glycose 5 mM, representing normoglycemic-NG conditions, with medium changed every two to three days. Cells were maintained in an incubator at 37 °C in a humidified (99%) atmosphere of 5% CO_2_ and 95% air. After the third passage, hDPCs were put into 6-well plates, and, at the confluence of about 80%, cells were subjected to high-glucose (25 mM) growth media for 72 h (hyperglycemic-HG conditions). Afterwards, NG and HG cells were exposed to MEL (0.1 mM or 1.0 mM, for 24 h), then collected, counted and stored. Total protein quantification was performed using Bradford assay. In the collected supernatants concentration of GDNF (pg/mL), iNOS and SOD were measured using the same commercial kit as for experiments on tissues.

### 2.4. Human Parotid Gland Cells (hPGCs) Culture

Mixed hPGCs cultures were established from two healthy non-diabetic (ND) and two type 2 diabetic participants. Excised tissue, obtained during surgical removal of the benign parotid tumor, was transported to the laboratory in media consisted of: low-glucose Dulbecco’s modified Eagle medium (DMEM), 20% heat-inactivated fetal bovine serum, penicillin-G (100 units/mL) and streptomycin (100 μg/mL). After gentle removal of all visible vessels and connective tissue, isolation of hPGCs from remaining tissue was performed by outgrowth method: tissue dissection with scissor and a blade into small pieces (0.5 mm^3^) and transferring them to the cell culture vials with addition of cell growth media (low-glucose DMEM, 10% heat-inactivated fetal bovine serum, 100 units/mL penicillin-G, 100 μg/mL streptomycin), which was changed every 3–4 days. The use of DMEM media supplemented with 10% serum was shown to allow long-term propagation of mixed salivary gland cells culture consisting of both epithelial and mesenchymal cells, able to proliferate and grow beyond passage 10 without alterations in morphology and proliferation rate [20]. Cells were maintained in an incubator at 37 °C with humidified atmosphere and with 5% CO_2_. When confluence of 80% was reached, cells were passaged using TrypLE Express. For cell counting and assessment of viability, trypan blue exclusion assay was performed. After the third passage, cells were put in 96-well plates for evaluation of melatonin cytotoxicity and in six-well plates for melatonin stimulation. According to in vitro cytotoxicity assays: 3-(4,5-dimethylthiazol-2-yl)-2,5-diphenyltetrazoliumbromide (MTT) and neutral red uptake (NR) tests, conducted in hDPCs and hPGCs, at two pharmacological concentrations of melatonin (Appendix A), we decided to use both concentrations in studies with hDPCs and the lower one in the studies with hPGCs. Cells treated with solvent only served as a control (96% ethanol in final concentration of 0.025%). After 24 h, cells were collected, counted and exposed to three freeze–thaw cycles in order to extract intracellular proteins with subsequent centrifugation and supernatant collection. The protein expression of GDNF (pg/mL) and iNOS (ng/mL) were quantified by ELISA and SOD activity (%) by spectrophotometric assay using the same commercial kits as for experiments conducted on hDPCs.

### 2.5. Bioinformatic Analysis

Bioinformatic tools presently engaged contain detailed information on biological pathways and processes in humans from the primary literature, representing peer-reviewed databases of different signaling molecules and their roles and relations, and thus allow for finding functional relationships between different signaling molecules in general. ShinyGO (v.0.75) application was used [21] (Available online: http://bioinformatics.sdstate.edu/go/, accessed on 15 April 2023) to perform the Compartments, RegNetwork and Reactome analyses. To investigate which common transcription factors (TF) interact with the investigated genes, we chose the “TF.Target” gene sets from RegNetwork [22]. The Compartments resource unifies complementary evidence on protein subcellular localization from curated knowledge, high-throughput experiments, text mining and sequence-based prediction methods [23]. Reactome is a freely available knowledgebase that consists of manually curated molecular events (reactions) organized into cellular pathways (https://reactome.org/, accessed on 15 April 2023) [24].

### 2.6. Statistical Analyses

Statistical analyses were performed in the GraphPad Prism 9 software (Graph Pad Software Inc., San Diego, CA, USA). The results were presented as mean  ±  SD. Chi-squared test was applied for testing categorical variables. Considering the small, random samples we had, as well as our aim to examine whether a diabetes or melatonin treatment had the effect, we used Student’s *t* test (for comparing means when the data are continuous and approximately normally distributed) and Mann–Whitney test (when the Student’s *t* test assumptions are not met). Data from cell cultures experiments were analyzed by Mann–Whitney test, and *p* < 0.05 was considered statistically significant.

A hypergeometric statistical test was applied to the GO annotations for investigated genes/proteins and functions with a false Discovery rate (FDR)-adjusted *p*-value  <  0.05 were considered significantly overrepresented. The top 10 pathways that had an adjusted *p*-value  <  0.05 were labeled as statistically significant and were reported in the results. For FDR calculation, the nominal *p*-value from the hypergeometric test was used. Fold enrichment is calculated as the percentage of proteins belonging to a pathway, divided by the corresponding percentage in the background.

## 3. Results

### 3.1. Study Group Characteristics

Demographic characteristics of study participants (parotid gland and dental pulp tissue donors) are presented in Table 1. There were no statistical differences in terms of age, gender, smoking or body mass index between non-diabetic and diabetic participants (Table 1).

### 3.2. Expression Levels of MEL, GDNF and iNOS in Parotid Gland and Dental Pulp Tissues of T2D Participants

Decreased MEL and increased GDNF protein expression levels were obtained in diabetic vs. non-diabetic parotid gland (GDNF: 186.9 ± 21.7 vs. 113.1 ± 11.6, *p* < 0.01; MEL: 202.3 ± 19.1 vs. 270.2 ± 11.7, *p* < 0.01; Student’s *t* test, Figure 2A), as well as in diabetic dental pulp tissue (GDNF: 33.0 ± 4.2 vs. 25.3 ± 2.5, *p* < 0.05; MEL: 124.9 ± 5.4 vs. 227.3 ± 10.3, *p* < 0.01; Student’s *t* test, Figure 2B). Associated with this, enhanced iNOS expression was observed in diabetic compared to non-diabetic parotid and pulp tissue (*p* < 0.01, Figure 2A,B).

### 3.3. Pearson’s Correlation Coefficient

Strong negative association has been found between MEL and GDNF, both in non-diabetic and diabetic: dental pulp (r = −0.84, *p* < 0.001 in non-diabetic and r = −0.94, *p* < 0.001 in diabetic) and parotid gland (r = −0.68, *p* < 0.01 in non-diabetic and r = −0.89, *p* < 0.001 in diabetic). Significant negative association was found between MEL and iNOS in diabetic parotid gland (r = −0.82, *p* < 0.001) and pulp (r = −0.85, *p* < 0.001), while significant positive association was found between GDNF and iNOS in diabetic parotid gland (r = 0.77, *p* < 0.001) and pulp (r = 0.79, *p* < 0.001)

### 3.4. GDNF Expression in hDPCs Treated with MEL

Hyperglycemic cells express higher GDNF levels compared to normoglycemic hDPCs. The addition of MEL1 (0.1 mM) increased GDNF in normoglycemic hDPCs, while MEL2 (1 mM) decreased GDNF in hyperglycemic hDPCs (Figure 2C).

### 3.5. GDNF, iNOS Expression and SOD Activity in hPGCs Treated with MEL

Higher levels of GDNF, iNOS and SOD activity were shown in diabetic compared to non-diabetic hPGCs. Addition of MEL (0.1 mM) increased GDNF and SOD levels in non-diabetic hDPCs and decreased iNOS and SOD levels in diabetic hDPCs (Figure 3)

### 3.6. Enrichment Analyses

The regulation of melatonin synthesis primarily depends on the expression level and enzyme activity of arylalkylamine N-acetyltransferase (AANAT), the rate-limiting enzyme in melatonin biosynthesis [25]. The network analysis of GDNF and AANAT genes detected top-ranked significant transcriptional factors: ATF 1-7, CREB1 and HIF-1-beta as the key transcriptional regulatory factors (Table 2).

Reactome Knowledgebase was used for an enrichment analysis of biological pathway involving proteins engaged in T2DM-associated alterations in human parotid gland and dental pulp, and the 10 most significantly enriched pathways are shown in Figure 4A. These were pathways related to the immune system and cellular stress responses. Four out of five proteins, GDNF, iNOS, SOD1 and SOD2, were found in the cytokine signaling in the immune system and signaling by interleukins signaling. Jensen COMPARTMENTS also demonstrated that investigated proteins, GDNF, iNOS, SOD1 and SOD2 are co-expressed with melatonin receptor 1A (MTNR1A), prominently in mitochondrial permeability transition pore complex (Figure 4B).

## 4. Discussion

Recent studies revealed neurotrophin signaling as critical for targeting salivary dysfunction [26,27]. Namely, Xiao et al. established that neurotrophin, GDNF, may stimulate murine salivary stem cell growth as well as could improve saliva production and enrich the number of functional acini in submandibular glands [27]. GDNF was also shown to promote survival and proliferation of dental pulp cells [28]. Since MEL showed the ability to promote the production of GDNF in the nervous system [29], we investigated whether GDNF is involved in stress-protective effects of MEL in diabetic glands and dental pulp.

The present study revealed for the first time that patients with T2DM express increased GDNF and decreased MEL content, in both parotid gland and dental pulp tissues, which show strong negative correlation under the state of oxidative–nitrosative stress. Decreased glandular MEL content could be related to a lower number of melatonin granules observed by Isola et al. [30] in human diabetic parotid and submandibular glands, thereby contributing to altered secretory activity of glands. Up-regulated GDNF in the diabetic tissues could reflect diabetes-enhanced protein glycosylation [31], a process required for correct processing of the GDNF protein, but also closely linked to oxidative stress. Namely, GDNF exerts a variety of antioxidant/anti-inflammatory effects, such as reducing oxidative stress via transcriptional regulation of glutathione synthesis, attenuation of programmed cell death, and a decrease in the inflammatory response via modification of the iNOS expression [32,33]. Moreover, GDNF increase was reported to represent a neuroprotective strategy in the state of oxidative stress [34]. Our results show that diabetes induces oxidative–nitrosative stress in the diabetic parotid gland as well as in diabetic dental pulp [3], reflected by increased iNOS expression and enhanced total SOD activity, and that accompanied GDNF upregulation may be a mutual nitrosative stress-responsive mechanism, given the strong positive correlation between iNOS and GDNF presently observed under diabetic conditions.

In order to investigate the functional relationship between MEL and GDNF under diabetic conditions, hyperglycemic hDPCs and diabetic hPGCs were used. The results showed that MEL (0.1 mM) per se increased GDNF in normoglycemic dental pulp cells and non-diabetic hPGCs, which is in line with reports of up-regulation of GDNF expression via melatonin receptor signaling by physiological and pharmacological MEL concentrations in neural components in vitro [35] and in vivo conditions on animal models [36]. However, in hyperglycemic hDPCs and diabetic hPGCs, melatonin (0.1 mM) did not increase GDNF expression, while at high pharmacological concentration (1 mM), MEL reduced GDNF in hDPCs. In the light of the present and our previous results of high melatonin-induced down-regulation of iNOS and SOD activity in hyperglycemic hDPCs [3], observed down-regulation of GDNF by MEL could reflect overall alleviation of oxidative–nitrosative stress. The latter is supported by the present results showing a strong positive association between iNOS and GDNF, as well as a negative association between MEL and iNOS, as well as MEL and GDNF in diabetic tissues of dental pulp and parotid gland. The complexity of MEL action seems to be pronounced by the present results showing different MEL effects on GDNF expression under non-diabetic and diabetic conditions, suggesting MEL acts via seemingly opposite mechanisms under pathological and physiological conditions, as has been proposed previously [37]. The observed difference in the concentration-dependent MEL effect in hDPCs may be related to the high-dose MEL property to reduce the degree of oxidative stress-triggered protein glycosylation [38], which could explain the observed decrease in GDNF expression in vitro, in the presence of both hyperglycemia and melatonin. Having in mind the significance of the glyco-redox interplay in inflammation in diabetes [39], the observed MEL effect represents a so far unrecognized aspect of antioxidant, anti-inflammatory and diabetes-protective effect of MEL. Indeed, bioinformatic analysis revealed that GDNF-MEL protein interaction occurs at the level of the ATF/CREB family of transcription factors involved in regulation of inflammation and oxidative stress responses [40,41,42]. Besides glycosylation, the delicate balance of the redox potential is tightly regulated by the mitochondrial function. Furthermore, diabetes mellitus is closely related to alterations in the function of mitochondria, such as in terms of the pathophysiological event called the opening of the mitochondrial permeability transition pore, which induces proapoptotic proteins from mitochondria release, and resulting in cell death [43]. Bioinformatic analysis showed that all investigated proteins (GDNF, iNOS, SOD isoforms) share subcellular localization in the mitochondrial permeability transition pore complex, along with major melatonin receptor MNTR1A protein, pointing to their importance for the observed effects in diabetes. Furthermore, Reactome pathway enrichment analysis revealed that GDNF, iNOS and SOD isoforms are significantly involved in detoxification of reactive oxygen species and cytokine signaling pathway/signaling by interleukins, suggesting their involvement in oxidative, inflammatory and immune-mediated pathophysiological processes underlying T2DM.

## 5. Conclusions

As far as we know, this is the first study showing the involvement of neurotrophic factor GDNF in melatonin cellular action in oral tissues in type 2 diabetes, via predicted stress-response and ATF/CREB transcription factors signaling pathways. MEL induced stress-protective mechanism in hyperglycemic hDPCs and diabetic hPGCs, suggesting MEL could be beneficial for diabetes- associated disturbances in oral tissues.

## Figures and Tables

**Figure 1 ijerph-20-06727-f001:**
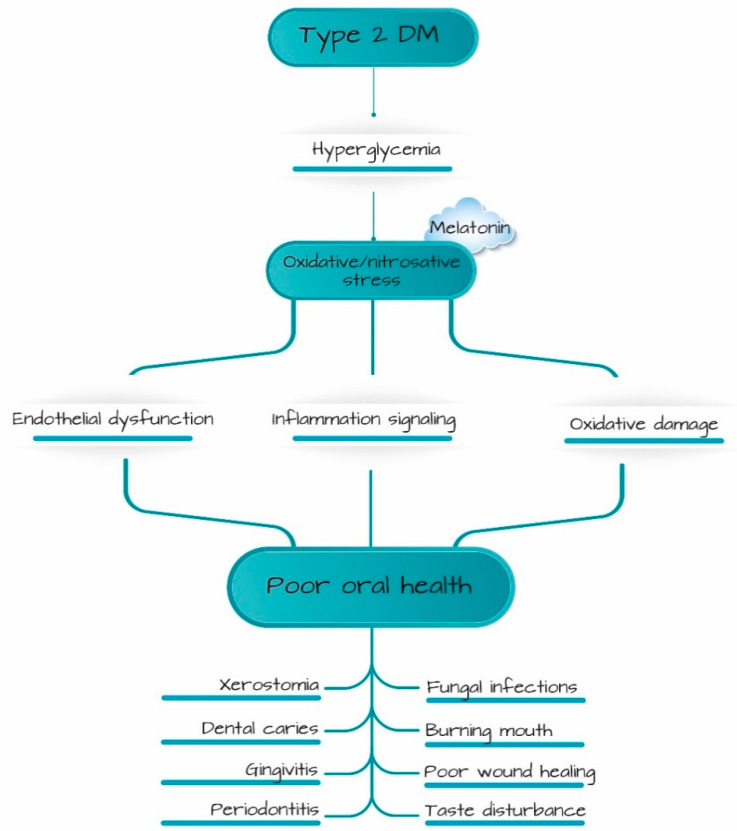
Conditions associated with poor oral health due to type 2 diabetes-induced deterioration of function of salivary gland and dental pulp. Melatonin may be protective in oral tissues in diabetes by targeting oxidative–nitrosative stress.

**Figure 2 ijerph-20-06727-f002:**
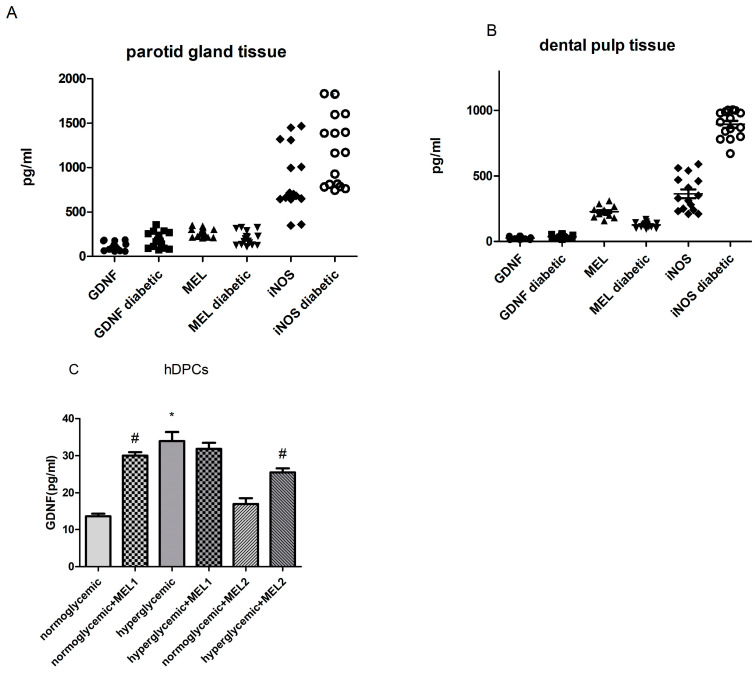
Expression levels of different proteins in human non-diabetic and diabetic parotid gland (**A**) and dental pulp (**B**) as well as effects of different concentrations of MEL on GDNF protein expression levels in normoglycemic and hyperglycemic human dental pulp cells (**C**). The graphs (**A**,**B**) represent individual measurements of expression levels of MEL, GDNF and iNOS in parotid gland and dental pulp. Box plots (**C**) represent mean ± SD. * *p* < 0.05 normoglycemic compared to hyperglycemic cells, Mann–Whitney test # *p* < 0.05 cells with compared to without MEL. Mann–Whitney test; MEL1 = 0.1 mM; MEL2 = 1 mM.

**Figure 3 ijerph-20-06727-f003:**
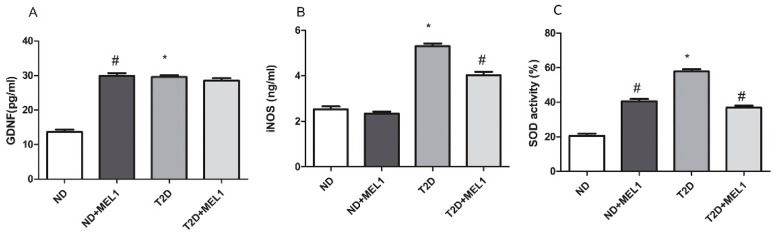
Effects of melatonin on expression levels of different proteins in human non-diabetic and diabetic parotid glands. The box plot graphic represents the mean ± SD of expression levels of GDNF (**A**) and iNOS (**B**) protein expression and SOD activity levels (**C**) in non-diabetic (ND) and type 2 diabetic (T2D) human parotid gland cells. * *p* < 0.05 nondiabetic compared to diabetic, Mann–Whitney test; # *p* < 0.05 cells with compared to without MEL, Mann–Whitney test; MEL1 = 0.1 mM.

**Figure 4 ijerph-20-06727-f004:**
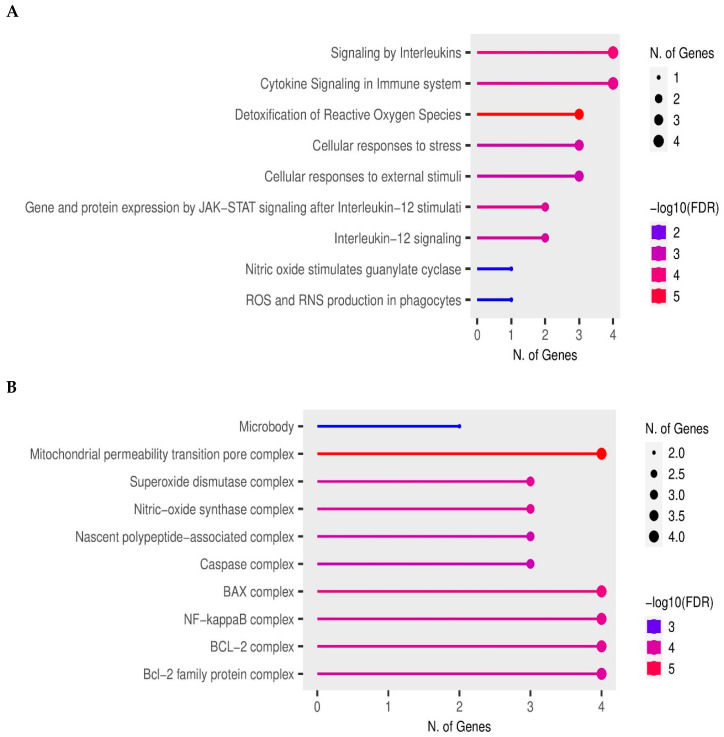
Reactome pathways enrichment analysis for proteins involved in type 2 diabetic stress-responses in human parotid gland and dental pulp (**A**) and Jensen COMPARTMENTS enrichment analysis representing associations between proteins involved in type 2 diabetic stress-responses in human parotid gland and dental pulp with cellular compartments (**B**). The number of proteins enriched in each pathway/compartment is presented on the x-axis, and the top 10 most significantly enriched pathways/compartments involving the investigated proteins are presented on the y-axis.

**Table 1 ijerph-20-06727-t001:** Characteristics of nondiabetic and type 2 diabetic study participants.

Characteristics	Nondiabetic *n* = 31	Type 2 Diabetic *n* = 31
Woman/man ^†^	15/16	16/15
Age (years, mean ± SD) ^‡^	59.1 ± 11.6	67.1 ± 9.7
Smokers (%) ^†^	37.5	31.25
BMI (kg/m^2^, mean ± SD) ^§^	23.4 ± 2.8	26.1 ± 3.3
Disease duration (years, mean ± SD)		6.1 ± 2.8

*n*—number of participants; *p* > 0.05 (^†^ Chi-square test; ^‡^ Mann–Whitney test; ^§^ Student’s *t*-test for independent samples).

**Table 2 ijerph-20-06727-t002:** The coregulatory transcription factors for studying GDNF and MEL (AANAT) genes’ regulatory systems *.

Genes	Fold Enrichment	Pathways
GDNF, AANAT	39.4	ATF5 target gene
GDNF, AANAT	39.3	ATF7 target gene
GDNF, AANAT	39.0	ATF3 target gene
GDNF, AANAT	36.8	ATF4 target gene
GDNF, AANAT	14.9	CREB1 target gene
GDNF, AANAT	13.0	HIF-1-beta target gene

* TF. Target gene sets from RegNetwork. As a measure of effect size, fold enrichment indicates how drastically genes of a certain pathway are overrepresented.

## Data Availability

The data presented in this study are available on request from the corresponding author.

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
