# Peer review of "Melatonin Action in Type 2 Diabetic Parotid Gland and Dental Pulp: In Vitro and Bioinformatic Findings"

_ijerph, 2023, doi:10.3390/ijerph20186727_

Round 1

Reviewer 1 Report

The article is interesting since it pretends to explain the mechanism of action of melatonin parotid glands and dental pulps in patients with diabetes.

The measurements of iNOS, GDNF, and melatonin from human sample seems to be consistent, however bar plots should be replaced with plot of individual values to better appreciate the dispersion. Also each measurement should be presented in individual graphs to better equalize the scales.

The panel C of the figure 2 is the most intriguing of all the results, and make the conclusion not to be supported by this results. More experiments with a greater range of doses should we needed to clarify this point especially using lower doses of melatonin. One milimollar is already a huge dose.

Furthermore,  in figure 3 A, melatonin doesn't seem to modify the levels of GDNF, which is also not consistent with the conclusion. Panels B and C add little to the already established antioxidant action of melatonin.

It is not clear to me if the Reactome knowledgebase and the Jensen compartments are specifics for dental pump and parody glands or if this pathways are modified in general, please clarify.

Author Response

Thank you for the suggestions that significantly improved our paper.

Accordingly, we modified the graphs in order to better appreciate the dispersion of the individual measurements, although, due to concise reporting, we decided to use a single graph for measurements in the dental pulp and single for measurements in the parotid gland.

According to MTT and NR tests conducted in hDPCs and hPGCs at two pharmacological concentrations of melatonin (supplemental file added), we decided to use both concentrations in studies with hDPCs and the lower one in the studies with hPGCs. We decided to use high MEL concentrations since MEL exhibit dose-dependent free radical scavenging properties, at the same time showing lack of toxicity (Tan DX et al., 1993). Having this in mind, and results obtained in cells deriving from both, dental pulp and parotid gland, 0.1mM of MEL increases GDNF in nondiabetic tissues while show no effects under diabetic conditions. However, 1mM MEL normalized GDNF levels in hyperglycemic hDPCs, what could reflect decrease in hyperglycemia-induced stress since strong positive correlation was obtained between iNOS and GDNF as well as negative correlations between MEL and iNOS and MEL and GDNF in diabetic dental pulp and parotid gland tissues. Moreover, our previous studies in hyperglycemic hDPCs showed that 1mM MEL and not 0.1mM MEL normalized hyperglycemia-induced iNOS levels. The complexity of MEL action seems to be pronounced by present results showing different MEL effects on GDNF expression under non-diabetic and diabetic conditions, suggesting MEL acts via seemingly opposite mechanisms under pathological and physiological conditions, as has been proposed previously ( https://doi.org/10.1007/s10456-019-09689-7). Thus, we added the sentences in Discussion in order to clarify observed results and modified Conclusions.

Bioinformatic tools engaged presently collect detailed information on biological pathways and processes in humans from the primary literature and represent peer-reviewed databases of different signaling molecules, their roles and relations, thus allow finding functional relationships between different signaling molecules in general.

Reviewer 2 Report

In the current study, the authors have reported that melatonin action in type 2 diabetic parotid gland and dental pulp in the in vitro study. It is a sound study, but several issues should be addressed.

1.       The most important observation presented in the current study was the association between melatonin and GDNF. Since the authors have not explored the potential molecular connections between them it is difficult to believe that they have the cause-result effect, in other word, this association may be just a coincidence.  

2.       The results were also not logically consistent. As mentioned by authors that the decreased melatonin was associated with increased GDNF. However, this was not the case in the results presented in figure 3 A. In the ND plus melatonin group, the GDNF was even higher the its control (ND) group, in addition, the GDNF was still high in the T2D plus melatonin group. These inconsistences were not discussed in the Discussion.

3.       In Figure 2, the unit of iNOS (pg) was 1000 times difference with the unit (ng) described in the Methods.

4.       The unit of the SOD activity, %, should be compared to a standard. This has not detailed in the text.

5.       The English should be improved  by a native English speaker

Need improvement!

Author Response

Thank you for the suggestions that significantly improved our paper.

  1. It is correct that present results do not support cause-result association between MEL and GDNF. However, based on the present association (correlation) between GDNF and MEL obtain in dental pulp and parotid gland, as well as bioinformatic analyses showing relations between MEL and GDNF and their predicted associated engagement in stress-related signaling pathways, we believe that this association is more than just coincidence. Nevertheless, in order to clarify more the complex relations: MEL-GDNF-oxidative stress, we modified Discussion and Conclusions.
  2. According to MTT and NR tests conducted in hDPCs and hPGCs at two pharmacological concentrations of melatonin (supplemental file added), we decided to use both concentrations in studies with hDPCs and the lower one in the studies with hPGCs. We decided to use high MEL concentrations since MEL exhibit dose-dependent free radical scavenging properties, at the same time showing lack of toxicity (Tan DX et al., 1993). Having this in mind, and results obtained in cells deriving from both, dental pulp and parotid gland, 0.1mM of MEL increases GDNF in nondiabetic tissues while show no effects under diabetic conditions. However, 1mM MEL normalized GDNF levels in hyperglycemic hDPCs, what could reflect decrease in hyperglycemia-induced stress since strong positive correlation was obtained between iNOS and GDNF as well as negative correlations between MEL and iNOS and MEL and GDNF in diabetic dental pulp and parotid gland tissues. Moreover, our previous studies in hyperglycemic hDPCs showed that 1mM MEL and not 0.1mM MEL normalized hyperglycemia-induced iNOS levels. The complexity of MEL action seems to be pronounced by present results showing different MEL effects on GDNF expression under non-diabetic and diabetic conditions, suggesting MEL acts via seemingly opposite mechanisms under pathological and physiological conditions, as has been proposed previously (https://doi.org/10.1007/s10456-019-09689-7). Thus, we added the sentences in Discussion in order to clarify observed results and modified Conclusions.

  1. According to manufacturer’s instructions, measured iNOS was expressed in ng/ml. Indeed, we obtain iNOS in ng/ml, but in order to present all results on one graph, we expressed iNOS levels in pg/ml. (0.3 ng= 300 pg).
  2. Superoxide Dismutase Activity Assay Kit is colorimetric test based on reaction of superoxide anions on WST-1 to produce a water-soluble formazan dye which can be detected by the increase in absorbance at 450 nm. The kit can be used without a standard since it is reporting as %inhibition. The greater the activity of SOD in the sample, the less formazan dye is produced.  Calculation was made according to equation:

SOD Activity (inhibition rate %) = (Ablank1 – Ablank3) – (Asample – Ablank2) X 100/ (Ablank1 – Ablank3), where A = absorbance

  1. Thank you. The manuscript has been checked by a collegue, native English speaker (Mr. A. R.).

Reviewer 3 Report

Methodological Biases exist

(The Authors must see my remarks)

Author Response

Thank you for the suggestions that significantly improved our paper.

We made all spelling corrections required and manuscript was read by a colleague, native English speaker (Mr. A. R.).

The aim was to Figure 1 represent clinical features of diabetes-induced salivary gland and dental pulp deterioration, and why melatonin may act protectively in both tissues. In order to clarify it, we modified the Figure title.

We omitted trade names except for ELISA commercial kits.

We added exclusion criteria and modified the paragraph in Methodology.

The present study investigates melatonin cellular effects in dental pulp and parotid gland and influence of diabetes on these effects. Considering a small, random samples we had, and we aimed to compare the means of groups with or without melatonin treatment, or the means of two different populations, non-diabetic and diabetic, we used Student t test (for comparing means when the data are continuous and approximately normally distributed) and Mann-Whitney test (when the t-test assumptions are not met). Since we aimed to determine whether a diabetes or melatonin treatment had the effect, and not for prediction purposes, we did not used regression models.

We have presented just baseline characteristics of participants we had easily available, without other hardly feasible specific clinical measurements, since experiment put focus on basic cellular rather than clinical features of diabetes.

We made suggested alterations in paragraph showing the results of Pearson’s correlation coefficient.

We made corrections in the Figure legends by adding data of statistical test used.

The coregulatory transcription factors for studying GDNF and MEL (AANAT) genes regulatory systems originates from RegNetwork. Bioinformatic tools engaged presently collect detailed information on biological pathways and processes in humans from the primary literature and represent peer-reviewed databases of different signaling molecules, their roles and relations, thus allow finding functional relationships between different signaling molecules in general. References of the bioinformatic tools we used were given in the Methodology section.

Round 2

Reviewer 2 Report

The quality of the revised version is improved. The English needs  further improvement. 

With minor English polish, The manuscript is acceptable.

Author Response

Thank you for the remark. The manuscript is revised according to suggestions by native English speaker, Mr. Andrew Roberts.
